# Integrative Analysis of Regulatory Module Reveals Associations of Microgravity with Dysfunctions of Multi-body Systems and Tumorigenesis

**DOI:** 10.3390/ijms21207585

**Published:** 2020-10-14

**Authors:** Mengqin Yuan, Haizhou Liu, Shunheng Zhou, Xu Zhou, Yu-e Huang, Fei Hou, Wei Jiang

**Affiliations:** Department of Biomedical Engineering, Nanjing University of Aeronautics and Astronautics, Nanjing 211106, China; bioyuanmengq@gmail.com (M.Y.); liuhaizhou@nuaa.edu.cn (H.L.); zhoushunheng@nuaa.edu.cn (S.Z.); zhouxu817@nuaa.edu.cn (X.Z.); yuehuang@nuaa.edu.cn (Y.-e.H.); houfei@nuaa.edu.cn (F.H.)

**Keywords:** microgravity, feed-forward loop, drug, miRNA, regulatory network, cancer

## Abstract

Previous studies have demonstrated that microgravity could lead to health risks. The investigation of the molecular mechanisms from the aspect of systems biology has not been performed yet. Here, we integratively analyzed transcriptional and post-transcriptional regulations based on gene and miRNA expression profiles in human peripheral blood lymphocytes cultured in modeled microgravity. Two hundred and thirty dysregulated TF-miRNA (transcription factor and microRNA) feed-forward loops (FFLs) were identified in microgravity. The immune, cardiovascular, endocrine, nervous and skeletal system subnetworks were constructed according to the functions of dysregulated FFLs. Taking the skeletal system as an example, most of genes and miRNAs in the subnetwork were involved in bone loss. In addition, several drugs have been predicted to have potential to reduce bone loss, such as traditional Chinese medicines Emodin and Ginsenoside Rh2. Furthermore, we investigated the relationships between microgravity and 20 cancer types, and found that most of cancers might be promoted by microgravity. For example, rectum adenocarcinoma (READ) might be induced by microgravity through reducing antigen presentation and suppressing IgA-antibody-secreting cells’ migration. Collectively, TF-miRNA FFL might provide a novel mechanism to elucidate the changes induced by microgravity, serve as drug targets to relieve microgravity effects, and give new insights to explore the relationships between microgravity and cancers.

## 1. Introduction

The National Aeronautics and Space Administration (NASA) twins study indicated that space flight could induce a series of physiological and pathological changes in astronauts, including immunological stress, vascular changes, and bone loss [1]. Hitherto, some ground-based machines are used to simulate microgravity, such as clinostats and rotating wall vessel bioreactors [2,3]. Simulated microgravity is convenient for us to understand the effects of real microgravity better. During the last three decades, numerous studies have focused on molecular mechanisms, of which microgravity could lead to human body system dysfunctions. For example, the increased RANKL/OPG ratio due to microgravity induced osteoclastgenesis and bone resorption [4]. For the immune system in microgravity, the significant reduction of IL2 and IL2 receptor alpha expression and downregulated PAK pathway prevented the full activation of T-cell [5,6].

Transcription factor (TF) and microRNA (miRNA) are two types of well-studied gene regulators. TFs control the transcription of DNA to messenger RNA (mRNA) through binding to promoter. It is well known that the dysregulation of TFs can influence various biological effects such as stress and adaptive responses [7]. MiRNAs (~22 nucleotides) inhibits the translation of target mRNA by degrading and silencing mRNA. Accumulating evidence has shown that the alternations of miRNA expression in microgravity lead to metabolic or functional changes in cells [8]. TF and miRNA can coordinately regulate gene expression through forming feed-forward loops (FFLs), which has been proposed as an effective tool to explore the molecular mechanism of many diseases [9,10]. However, the regulatory mechanism about TF-miRNA FFLs in microgravity has not been explored yet.

In addition, several studies have focused on the relationships between microgravity and cancers. Some studies suggested that microgravity could induce the risk of cancer. For example, spaceflight can weaken astronauts’ immune system and cause a high risk of cancer. Early T-cell activation genes in *mouse*, such as *interferon 2*, *Tagap* and *ligp1*, were significantly downregulated in microgravity [11]. However, the inhibitory effects of microgravity on the viability and growth of cancer cells were also observed. The upregulation of p53 and bax was found to increase apoptosis in human follicular thyroid carcinoma cells due to microgravity [12]. The stemness loss of lung cancer stem cell was induced by the downregulation of *NANOG* and *OCT4* in microgravity [13]. However, there is no systematic research about the associations between microgravity and pan-cancer.

In this study, we systematically analyzed the effects of microgravity on five human body systems, as well as the associations between microgravity and pan-cancer. Firstly, we identified 230 dysregulated FFLs in microgravity. Then, human body system subnetworks were constructed according to the functions of FFLs, including immune, cardiovascular, skeletal, endocrine, and nervous system subnetworks. Moreover, almost all the genes and miRNAs in skeletal system subnetwork were associated with bone loss, which is one of the major risks for astronauts in space flight. Next, we predicted the candidate drugs Emodin and Ginsenoside Rh2, which could reverse the expression of dysregulated miRNAs, to prevent or reduce the bone loss due to microgravity. In addition, we investigated the relationships between microgravity and cancers. The results suggested that microgravity might promote most cancers. For example, microgravity promoted rectum adenocarcinoma (READ) through reducing antigen presentation and suppressing IgA-antibody-secreting cells’ migration to inhibit immune protection. In summary, we identified the dysregulated FFLs in microgravity, elucidated how microgravity influences the function of human body systems, screened potential drugs to inhibit bone loss in microgravity, and explored the relationships between microgravity and cancers.

## 2. Results

### 2.1. Characteristics of TF-miRNA Regulatory Network

We obtained the experimentally validated TF and miRNA regulations from the TRANSFAC [14], TarBase [15], miRTarbase [16] and TransmiR [17] databases. After eliminating all self-loops, we obtained 4693 TF-gene, 3930 TF-miRNA, 12,052 miRNA-gene regulations to construct the integrated regulatory network, which included 647 TFs, 888 miRNAs and 3794 genes (Appendix A). Then, we analyzed the topological properties of the network. Similar to most biological networks, the degree distribution of the background regulatory network approximately displayed a power law distribution (Appendix A), which meant that it satisfied scale-free topology [18].

### 2.2. Dysregulated FFLs in Microgravity

We identified 4064 FFLs in the TF-miRNA regulatory network. Furthermore, through integrative analysis of gene and miRNA expression profiles of human peripheral blood lymphocytes (HPBLs) incubated in microgravity and normal gravity, we obtained 230 dysregulated FFLs, including 128 TF-FFLs, 62 miRNA-FFLs, and 40 feed-back FFLs (FB-FFLs) (Figure 1A–C and Appendix A, details in Materials and Methods). By merging these FFLs, we constructed the dysregulated FFL network including 40 TFs, 67 miRNAs, and 67 genes.

The Kyoto Encyclopedia of Genes and Genomes (KEGG) pathway enrichment analysis of the genes and the miRNAs in the dysregulated FFLs was performed by Database for Annotation, Visualization and Integrated Discovery (DAVID) [19] and DIANA-miRPath [20], respectively. The results showed that some immune system-related pathways were significantly enriched, including hepatitis B, HTLV-1 infection and inflammatory bowel disease, which were consistent with the function of HPBLs (Figure 1D,E). Interestingly, many cancer-related pathways were also identified, such as pathways in cancer, viral carcinogenesis, and the PI3K-Akt signaling pathway (Figure 1D,E). 

Because the nodes with high connectivity usually play critical roles in the network, we calculated the degree of each node in the dysregulated FFL network. Then, we defined the FFLs in which the degrees of all nodes are no fewer than 10 as hub FFLs. As a result, we obtained 31 hub FFLs. Then, the hub FFL subnetwork was constructed through connecting the hub FFLs, which included 10 TFs, eight miRNAs and five genes (Figure 1F). In the hub FFL subnetwork, we found that most of the genes were involved in cancer progression. For example, *KLF4*, which is zinc-finger transcription factor, is a tumor suppressor in colorectal cancer and may act as oncogene in oral squamous cell carcinoma [21,22]. Protein encoded by *RELA* is a subunit of NFKB, which may have a potential role as prognostic biomarker in prostate cancer [23]. To further validate this phenomenon, 724 known cancer genes were retrieved from the Cancer Gene Census (CGC) database [24]. We found that eight (~53%) genes in the hub FFL subnetwork were known cancer genes. This proportion was significantly higher than that in the dysregulated FFL network (35%, Fisher’s exact test *p*-value = 0.035) and the background regulatory network (10%, Fisher’s exact test *p*-value < 0.01). Meanwhile, the proportion of the known cancer genes in dysregulated FFLs was significantly larger than background network (Fisher’s exact test *p*-value < 0.01) (Figure 1G). The results indicated that the genes in the dysregulated FFL network, especially in the hub FFL subnetwork, were more likely to be the cancer genes. We will further analyze the associations between microgravity and cancers in the pan-cancer cohort.

### 2.3. The Microgravity Effects on Multi-body Systems

The environment of microgravity will bring about remarkable multisystem changes in the human body, which lives in earth’s gravity. These changes cover a wide spectrum ranging from mild symptoms, such as nausea and transient sensory imbalances, to system function impairment [25]. Some studies have focused on the effect of microgravity on a single system including the cardiovascular system, skeletal system and so on. To explore the effect of microgravity on multiple human body systems including the immune, cardiovascular, skeletal, endocrine, and nervous systems, we performed KEGG functional annotation analysis of genes and miRNAs in dysregulated FFLs by DAVID and DIANA-miRPath, respectively. The dysregulated FFL related to one human body system was defined as that all nodes in the FFL should participate in the pathways related to this body system. We obtained 160 FFLs related to the functions of the five body systems, of which 133, 83, 46, 8 and 51 FFLs were identified for immune, cardiovascular, nervous, skeletal and endocrine systems, respectively, including 38 TFs, 31 genes and 46 miRNAs (Figure 2 and Appendix A). There were 67 system-specific FFLs, and 52 system-general FFLs, which were shared by at least three systems (Figure 2). 

In addition, we constructed five subnetworks through merging each system-related FFLs (Figure 3). There were 70 of 115 nodes (66.7%) and 183 of 324 edges (56.5%) shared by at least two subnetworks. The nodes and edges in immune subnetwork included most of those in the other four subnetworks (Figure 4A,B). To identify the important nodes in each subnetwork, we selected the top 10 nodes according to their degrees. Fourteen of 23 nodes (60.9%) were shared by at least two subnetworks, of which *RELA* and *FOS* were shared by five subnetworks. In addition, we used NCMine to extract functional modules in the five subnetworks [26]. Three modules were found to be shared by at least three subnetworks (Figure 4C and Appendix A). The results suggested that the dysregulated FFLs might have a systematic effect on human body, in which different body systems cooperate as a whole.

### 2.4. Effects of Microgravity on Bone Loss and Drug Screening

Taking the skeletal system as an example, we analyzed gene and miRNA expression in skeletal system subnetwork. On the one hand, we identified some significantly differentially expressed (DE) genes in this subnetwork such as *FOS* (log_2_FC = −3.765, *p*-value < 0.001) and *IFNG* (log_2_FC = 4.513, *p*-value = 0.012). On the other hand, one of the advantages of FFL method is that some nonsignificantly DE molecules can be also identified. For example, we identified nonsignificantly DE genes, including *RELA* (log_2_FC = 0.957, *p*-value = 0.033), *ESR11* (log_2_FC = −0.56, *p*-value = 0.021), *TP53* (log_2_FC = 0.642, *p*-value = 0.187), *PTK2* (log_2_FC = 0.585, *p*-value = 0.228), TNF (log_2_FC = −0.193, *p*-value = 0.681) and *BAX* (log_2_FC = 0.11, *p*-value = 0.463) as well as nonsignificantly DE miRNAs, including hsa-miR-221-3p (log_2_FC = 0.121, *p*-value = 0.817), hsa-miR-125a-5p (log_2_FC = 0.532, *p*-value = 0.145), hsa-miR-7-5p (log_2_FC = 0.373, *p*-value = 0.084) and hsa-miR-125b-5p (log_2_FC = 0.18, *p*-value = 0.7) in this subnetwork (Figure 4D). Furthermore, we investigated the functions of genes and miRNAs in this subnetwork from literatures. We found that the dysregulation of genes and miRNAs could lead to bone loss, which is one of the most common and well-known diseases for astronauts in space flight, by reducing osteoblast and increasing osteoclast. Bone loss is caused by the balance disruption between osteoblastic bone formation and osteoclastic bone resorption [27]. Previous studies have demonstrated that the reduction in osteoblast can lead to bone formation decrease [28]. Some genes and miRNAs in the subnetwork were found to be associated with the decrease of osteoblast. For example, *FOS* has been observed in a high level during the proliferative period of osteoblast development [29]. *IFNG* could inhibit the proliferation of human osteoblastic SaOS-2 cells [30]. *PTK2* was shown to inhibit adipose-derived stem cell differentiation to osteoblast [31]. Decreased *BAX* expression could reduce apoptosis of osteoblast [32]. Hsa-miR-221-3p was demonstrated to be involved in osteoblast differentiation [33]. Hsa-miR-125b-5p inhibited osteoblastogenesis by regulate the expression of *ErbB2* [34]. Meanwhile, bone resorption is enhanced due to an increase in osteoclast [35]. Some genes and miRNAs in the subnetwork were found to be related to the increase of osteoclast. For example, *RELA* is a subunit of *NFKB*, whose reduction in expression has been proposed to be an effective approach to inhibit osteoclast formation and bone resorptive activity [36]. Hsa-miR-125a-5p promoted osteoclast differentiation through inhibiting *TNFRSF1B* protein expression [37]. *ESR1* can induce osteoclast apoptosis [38]. Taken together, the dysregulation of genes and miRNAs in skeletal system subnetwork might lead to bone loss by reducing osteoblast and increasing osteoclast.

In addition, to relieve the effects of microgravity on bone loss, we screened drugs to reverse the expression of miRNAs and genes in the subnetwork. Many studies have demonstrated that small molecules could modulate miRNA expression. We obtained the drugs that can disrupt miRNA expression from the SM2miR database, which collects the experimentally validated influences of small molecules on miRNA expression [39]. Because the four miRNAs, including hsa-miR-221-3p, hsa-miR-125b-5p, hsa-miR-125a-5p and hsa-miR-7-5p, were elevated in microgravity, we selected the drugs which can inhibit the expression of miRNAs in the skeletal system subnetwork as much as possible (Appendix A). Interestingly, we found that two traditional Chinese medicines Emodin and Ginsenoside Rh2 could inhibit the expression of hsa-miR-221-3p and hsa-miR-125b-5p (Figure 4E). Emodin and Ginsenoside Rh2 have been used as herbal remedy for more than 2000 years [40,41]. In previous studies, Emodin and Ginsenoside Rh2 have been demonstrated to reduce the expression of *IFNG* [42,43] and *NFKB* [44,45], which participate in the process of bone formation [30,36]. Furthermore, the two drugs have been proposed to be candidates for osteoporosis treatment by inhibiting osteoclastogenesis [44,46]. Meanwhile, we screened drugs which could reverse the expression of genes in the skeletal system subnetwork. First, we downloaded the gene expression profiles under small molecule perturbations in Cmap [47]. Then, DE genes were obtained for each instance according to the previous study [48]. Eight drugs could reverse at least three genes’ expression (Appendix A), but the associations with bone loss had not been reported. In conclusion, we screened some candidate drugs, such as Emodin and Ginsenoside Rh2, to alleviate the symptom of bone loss due to microgravity by reversing miRNA and gene expression.

### 2.5. Deep Insight into the Relationships Between Microgravity and Cancers

We further explore the relationships between microgravity and cancers based on gene expression data. The NASA twins’ study has shown that the expression of some genes related to immune function and DNA repair could not revert back to normal levels after returning from the space to earth [1]. The activity of immune cells was suppressed in microgravity, which increased the cancer risk for astronauts [49]. Thus, to guarantee the health of astronauts in space, it is urgent to systematically uncover the relationships between microgravity and cancers. We downloaded gene expression profiles and obtained significantly DE genes in 20 cancer types from The Cancer Genome Atlas (TCGA) project (Table 1). Next, we examined relative shift about the distribution of log_2_FC of expression values between significantly DE genes (upregulated and downregulated genes, respectively) in each cancer type and genes in microgravity by the Wilcoxon rank-sum test (details in Materials and Methods). We defined the significant up-top, down-bottom or both (*p*-value ≤ 0.1) as consistent patterns; the significant up-bottom, down-top or both (*p*-value ≤ 0.1) as reverse patterns. The result showed that 18 of 20 cancers (90%) had a significant down-bottom pattern (Figure 5A, Appendix A), which meant that downregulated genes in the 18 cancer types were preferred to be downregulated in microgravity. Meanwhile, five cancer types (HNSC, LUAD, CHOL, COAD and READ) had a significant up-top pattern, which meant that upregulated genes in the five cancer types were preferred to be upregulated in microgravity. These five cancer types showed consistent pattern with both down-bottom and up-top patterns, which suggested that microgravity might promote these cancers. Previous studies have reported that microgravity was favorable to cell growth of human colorectal carcinoma cell line [50]. The migratory ability of human lung cancer cell lines of adenocarcinoma was increased after exposing to microgravity [51]. There were studies showing that microgravity may induce some types of cancer including lung, liver, head and neck cancers [49]. In addition, six cancers had a significant up-bottom pattern. In these cancers, GBM is one of the most aggressive and fatal human brain cancers, which showed only one reverse up-bottom pattern. GBM cells have been found a decrease in cell proliferation and an increase in chemosensitivity to cisplatin in microgravity [52]. It suggested that microgravity might serve as an expectable role of protection for GBM patients. However, the other five cancers have conflict trend with significant up-bottom and down-bottom patterns, which suggested the complexity of relationships between microgravity and these cancer types. Furthermore, in order to check the reproducibility of our results, we analyzed the additional gene expression data of cancers by microarray (LUAD and GBM from GSE116959 and GSE19728, respectively). The same procedures and parameters were performed. The results showed that the downregulated genes in the LUAD were preferred to be downregulated in microgravity (*p*-value = 4.94 × 10^−18^), and the upregulated and downregulated genes in GBM were preferred to be reversed in microgravity (*p*-value = 0.011 and *p*-value = 0.029, respectively). These results indicated that microgravity might promote LUAD and inhibit GBM, which were consistent with the results from TCGA datasets (Appendix A).

To understand the mechanism of microgravity effect on cancers, we analyzed significantly disturbed pathways existing in both microgravity and cancers. Taking READ as an example, we firstly obtained the 559 DE genes in microgravity, of which 115 DE genes were DE in READ and showed same regulation directions with READ. Next, we performed KEGG pathway enrichment of 115 DE genes by DAVID. The top 10 significantly enriched pathways were almost related to the function of immune system (Figure 5B). The six downregulated genes in microgravity, *HLA-DPA1* (log_2_FC = −1.785, *p*-value = 0.002), *HLA-DQA1* (log_2_FC = −1.65, *p*-value = 0.003), *HLA-DPB1* (log_2_FC = −1.83, *p*-value = 0.0007), *HLA-DMB* (log_2_FC = −1.64, *p*-value = 0.002), *CXCL12* (log_2_FC = −1.52, *p*-value = 0.001) and *TNFSF13* (log_2_FC = −2.2, *p*-value = 0.001) (Figure 5C), were downregulated in READ and were enriched in “intestinal immune network for IgA production” pathway, which is well-known in colorectal cancer [53]. These molecules have been demonstrated to play important roles in cancer immunoediting [54]. The *HLA-DPA1*, *HLA-DQA1*, *HLA-DPB1* and *HLA-DMB* belong to major histocompatibility complex (MHC) class II (Figure 5D), whose deficiency inhibited the ability of antitumor and protective immunity by reducing antigen presentation [54]. The downregulated *TNFSF13* (also known as *APRIL*), a member of tumor necrosis factor ligand superfamily, inhibited B cell proliferation, maturation, and survival, which is crucial in antigen presentation [55]. The decreased *CXCL12*, produced by epithelial cells of colon, inhibited IgA antibody secreting cells’ migration into lamina propria of colon [56]. Our results suggested that microgravity might promote READ by reducing antigen presentation and suppressing IgA-antibody-secreting cells’ migration through “intestinal immune network for IgA production” pathway. *HLA-DPA1*, *HLA-DQA1*, *HLA-DPB1*, *HLA-DMB*, *TNFSF13* and *CXCL12* might be the biomarkers associated with the pathogenesis of READ promoted by microgravity.

## 3. Discussion

A better understanding of how microgravity influences health will benefit astronauts in the long-duration spaceflight. TFs and miRNAs as major regulators of gene expression at the transcriptional and post-transcriptional level could coordinately regulate the same target gene through forming TF-miRNA FFLs, which have been widely used to explore molecular mechanisms of development and progression in many cancer types. However, the systematic investigation of TF-miRNA FFLs in microgravity has not been performed yet. In addition, the relationships of microgravity and pan-cancer remain unclear.

This study is the first attempt to investigate the microgravity effects on multiple human body systems from the perspective of TF-miRNA FFLs. Here, we constructed five human body system subnetworks and explored the mechanism of microgravity effects on body systems. Taking skeletal system as an example, the dysregulation of most genes and miRNAs in the subnetwork leads to reduction of osteoblast and increase of osteoclast, which induced bone loss. In addition, we identified potential drugs to prevent bone loss through reversing the dysregulated expression of miRNAs and genes based on SM2miR and Cmap. For example, Emodin and Ginsenoside Rh2, which are traditional Chinese medicines, could inhibit the expression of hsa-miR-221-3p and hsa-miR-125b-5p. Interestingly, the previous studies had proposed the two drugs as candidates for osteoporosis treatment by inhibiting osteoclastogenesis [44,45]. Ellipticine, the top one in the predicted rank list from Cmap, could reverse the expression of four genes in the subnetwork, including *RELA*, *PTK2*, *BAX* and *FOS*. Although these predicted drugs in Cmap have not been reported to be involved in bone loss, they might serve as novel candidate sources for further experiment validation.

In addition, we found that the function of some genes and miRNAs in the dysregulated FFLs were associated with cancer. Also, the known cancer genes were significantly enriched in the hub FFLs. It suggested that, to some extent, microgravity might be correlated with cancer. We further explored the relationships between microgravity and cancers, the results demonstrated that microgravity might be a risk factor for 13 cancer types, whereas it might be a protective factor for GBM. Furthermore, we explored the mechanism of microgravity effect on cancers. *HLA-DPA1*, *HLA-DQA1*, *HLA-DPB1*, *HLA-DMB*, *CXCL12* and *TNFSF13*, downregulated both in microgravity and READ, were enriched in “intestinal immune network for IgA production” pathway. It has been demonstrated to play an important role in colorectal cancer [53]. Meanwhile, *HLA-DPA1*, *HLA-DQA1*, *HLA-DPB1* and *HLA-DMB* belong to MHC class II, one of the most important components in tumor-immune interactions. *TNFSF13* induces B cell proliferation, maturation, and survival. *CXCL12* contributes to the migration of IgA-antibody-secreting cells. These genes might be involved in the pathogenesis of READ promoted by microgravity through reducing antigen presentation and suppressing IgA-antibody-secreting cells’ migration [54,55,56]. 

Our results will benefit the astronauts’ physical in space flight. However, there are still some limitations in this study. Firstly, only five samples in microgravity were analyzed in this study due to the deficiency of microgravity data. A larger cohort of samples in microgravity with high-throughput sequencing data is needed in the future to understand microgravity effects more precisely. Secondly, the HPBLs were used to explore the effects of microgravity on body systems in this study, because the gene expression data of tissues in microgravity is still not available. Meanwhile, many studies had shown that the gene expression in peripheral blood can be served as the biomarkers of diagnosis, prognosis, and drug treatment for different cancers [57,58]. For example, *ANXA1* in peripheral blood mononuclear cells was found to be the diagnosis marker for solid tumors, such as breast cancer, lung cancer, and melanoma [57]. Although in the current situation, investigation of the effects of microgravity on multiple body systems from peripheral blood might be feasible to some extent, more omics data about human tissues in microgravity are still urgently needed. In addition, some studies observed the alternations of gene and miRNA expression under gravity changing, but the mechanisms remained to be addressed. As we know, the vestibular organs are sensors for gravity changes in vivo [59], but the mechanism about “how cells in vitro detect the gravitational changes” are still unclear. There were some hypotheses about it. The cytoskeletal element and mechanosensitive ion channels have been suggested to be the gravity receptor [60,61]. After exposure to real or simulated microgravity, the imbalance of adhesion and cytoskeleton has an effect on signaling cascades and downstream transcription events [62]. The activation of a mechanosensitive ion channel induced by gravitational force could lead to the activation or silencing of genes expression [61]. However, these hypotheses have not been confirmed by experimental data so far. Furthermore, more experiments should be used to validate the effectiveness of candidate drugs for bone loss in microgravity.

In summary, we identified the dysregulated TF-miRNA FFLs in microgravity, constructed five human body system subnetworks and explored the relationships between microgravity and cancers. Our analysis might provide novel insights into the roles of TF-miRNA FFLs in microgravity, will be helpful for drug screening to prevent the health risks of microgravity, and give a hint of associations between microgravity and cancers.

## 4. Materials and Methods

### 4.1. Data Collection and Processing

The gene and miRNA expression of HPBLs in microgravity and normal gravity were measured by microarray, which were performed by Girardi et al. [63]. They obtained the HPBLs from five healthy donors, which were incubated in normal gravity and modeled microgravity for 24 h, respectively. The modeled microgravity was simulated by the rotating wall vessel bioreactor. The gene and miRNA expression profiles were downloaded from the Gene Expression Omnibus (GEO) database (GSE57408 and GSE57400). More details about the design of experiment referred to [63]. The raw microarray data was normalized with Quantile normalization and then log2-transformed. We converted gene probe IDs to gene symbols. The average probe expression value was used as gene expression value if multiple probes corresponded to one gene. To unify the miRNA name, the miRNA name from miRbase V.10.1 was converted to miRbase V.22 using “miRNAmeConverter” R package [64]. Finally, the differential expression of genes and miRNAs was computed by the “limma” R package, which is based on Bayesian adjusted t-statistics from the linear models [65].

### 4.2. Construction of the TF-miRNA Regulatory Network

The regulations of TFs to genes were obtained from TRANSFAC Professional database (Release: 2014.2) [14]. The regulations of miRNAs to genes were obtained from TarBase v8 [15], miRTarbase [16] and TRANSFAC Professional database. In TarBase and miRTarbase, we only retained the miRNA regulations that have been validated by low-throughput experiments, such as Western blot and quantitative PCR (qPCR). The union of regulations in the three databases was used for the further study. The regulations of TFs to miRNAs were obtained from TransmiR v2.0 [17] and TRANSFAC Professional database. The union of the two databases was retained to construct the regulatory network. Next, we connected all regulatory pairs and eliminated all self-loops to constructed the integrated TF-miRNA regulatory network. Collectively, there were 647 TFs, 888 miRNAs and 3794 genes in the network (Table 2).

### 4.3. Identification of Dysregulated FFLs

The TF and miRNA coordinately regulatory FFL includes one TF, one miRNA, and one target gene. According to their regulations, the FFLs can be typically classified into three types: TF-FFL, miRNA-FFL, and FB-FFL. In a TF-FFL, TF regulates miRNA and gene at transcriptional level; miRNA represses gene expression at post-transcriptional level. In a miRNA-FFL, miRNA represses TF and gene; TF regulates gene. In a FB-FFL, TF and miRNA mutually regulate each other and both regulate the same target gene.

To evaluate the dysregulation of all FFLs, we considered the differential expression of all nodes and the differential coexpression of all edges in the FFL. Firstly, each node was scored according to the extent of differential expression using the following formula [9,66]:(1)$$S_{node}=\varphi^{-1}\left(1-2\times \left(1-\varphi \left(Diff_{node}\right)\right)\right)$$
(2)$$Diff_{node}=\left(-\log_{10}p_{i}\right)\times \left|\log_{2}FC\right|$$
where $\varphi^{-1}$ is the inverse normal cumulative distribution function, $p_{i}$ is the *p*-value which represents the significance of expression changes determined by the limma R package. *FC* is the fold change of this gene or miRNA expression. Next, each edge was scored according to the difference of gene coexpression between microgravity and normal samples using the following equations [9,66]:(3)$$S_{edge}=\varphi^{-1}\left(1-2\times \left(1-\varphi \left(\left|D\right|\right)\right)\right)$$
(4)$$D=\frac{F\left(r_{microgavity}\right)-F\left(r_{normal}\right)}{\sqrt{\frac{1.06}{n_{microgravity}-3}+\frac{1.06}{n_{normal}-3}}}$$
(5)$$F\left(r\right)=\frac{1}{2}\ln\frac{1+r}{1-r}$$
where $r_{microgavity}$ and $r_{normal}$ are the Spearman correlation coefficients of gene expression in microgravity and normal samples, respectively. Function of F is Fisher transformation. *n* is the number of samples. Finally, the FFL score is the weighted sum of node scores and edge scores as follows [9,66]:(6)$$s_{FFL}=\gamma \frac{\sum_{node\in FFL}S_{node}}{\sqrt{n_{node}}}+\left(1-\gamma \right)\frac{\sum_{egde\in FFL}S_{edge}}{\sqrt{n_{edge}}}$$
where $n_{node}$ and $n_{edge}$ are the numbers of nodes and edges in the FFL, respectively (here, both of them are 3). γ is weight parameter (0<γ<1), which controls the contribution of node score and edge score. Here, we considered that the weights of node and edge score were equally, and γ was set as 0.5.

Next, we performed permutation analysis to estimate the significance of each FFL score. Firstly, a random FFL was constructed by randomly selecting three molecules. This process was repeated 100,000 times. Secondly, the scores of the random FFLs were calculated through the above equations. Finally, the empirical *p*-value was defined as the proportion of random FFL scores larger than the real FFL score. In this study, the FFLs with *p*-value ≤ 0.05 were considered as the dysregulated FFLs.

### 4.4. Construction of the Dysregulated FFL Subnetworks Related to Human Body Systems

The KEGG functional annotation analysis of genes and miRNAs in dysregulated FFLs were performed by DAVID [19] and DIANA-miRPath [20], respectively. According to the functions of genes and miRNAs, we defined the system function-related FFL as that all nodes in the FFL should participate in the pathways related to the body system. Then, through connecting the dysregulated FFLs related to the same body system, we constructed the dysregulated FFL subnetworks for five body systems, including immune, cardiovascular, skeletal, endocrine, and nervous systems.

### 4.5. Differential Expression Analysis of 20 Cancer Types

The gene expression data of cancers was downloaded from TCGA RNA-seq data (level 3). After filtering cancer types with fewer than three cancer or normal samples, 20 cancer types were retained for further analysis. Next, the edgeR R package, which is based on negative binomial model, was used to calculate gene differential expression [67]. The trimmed mean of M values (TMM) method was used to normalize the count. Only genes with more than one count per million (CPM) in at least half of the samples were included in differential expression analysis. Finally, we obtained the significantly DE genes at the threshold of FDR ≤ 0.05 and |log_2_FC| ≥ 1.

### 4.6. Pan-Cancer Analysis of Associations Between Microgravity and Cancers

The genes were ranked by descending order based on log_2_FC values in microgravity. Then, the Wilcoxon rank-sum test was used to check whether DE genes (upregulated and downregulated genes, respectively) of each cancer type were primarily found at the top or bottom of the entire ranked list, respectively (up-top, up-bottom, down-top or down-bottom pattern). We defined the significant up-top, down-bottom or both (*p*-value ≤ 0.1) as consistent pattern; the significant up-bottom, down-top or both (*p*-value ≤ 0.1) as reverse pattern. The consistent patterns meant high coherence of gene expression between microgravity and cancer. The reverse patterns meant reverse tendency of gene expression between microgravity and cancer.

## Figures and Tables

**Figure 1 ijms-21-07585-f001:**
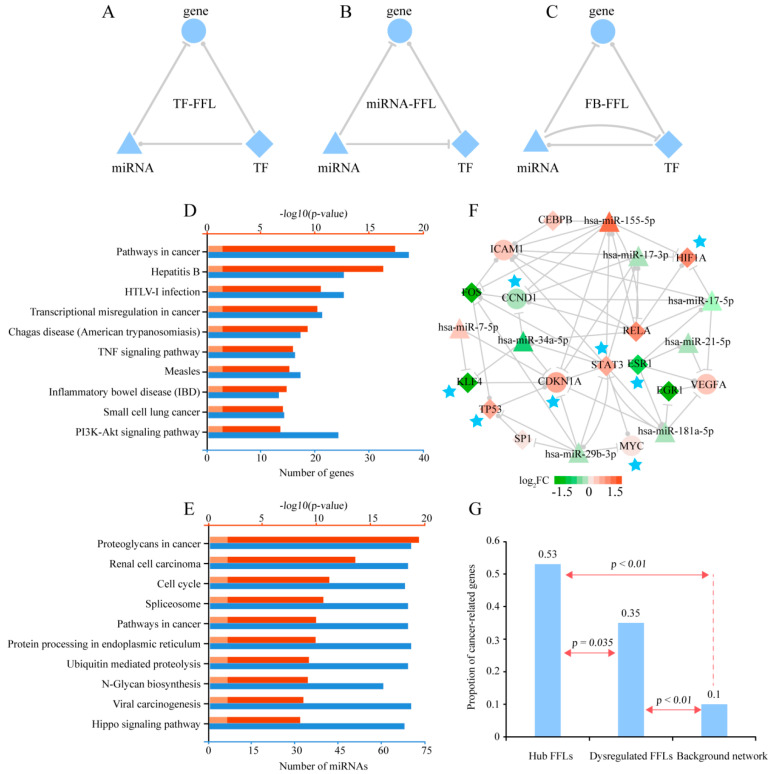
Three types of feed-forward loops (FFLs) and functional analysis of the dysregulated FFLs. (**A**). TF-FFL: transcription factor (TF) and microRNA (miRNA) regulate gene; TF regulates miRNA. (**B**). miRNA-FFL: TF and miRNA regulate gene; miRNA regulates TF. (**C**). FB-FFL (feed-back FFL): TF and miRNA regulate gene; TF and miRNA mutually regulate each other. Pathway enrichment analysis for genes (**D**) and miRNAs (**E**) in the dysregulated FFLs, respectively. The length of blue and red bar represents the number of genes (miRNAs) and –log10(*p*-value), respectively. The pink bar represents *p*-value = 0.05. (**F**). Dysregulated hub FFL subnetwork. Red and green represent upregulation and downregulation, respectively. The darker color represents larger |log_2_FC| value. The diamond, triangle and circular node represent TF, miRNA and gene, respectively. The blue stars indicate the known cancer genes. (**G**). Proportion of the known cancer genes in the hub FFL network, dysregulated FFL network and background regulatory network.

**Figure 2 ijms-21-07585-f002:**
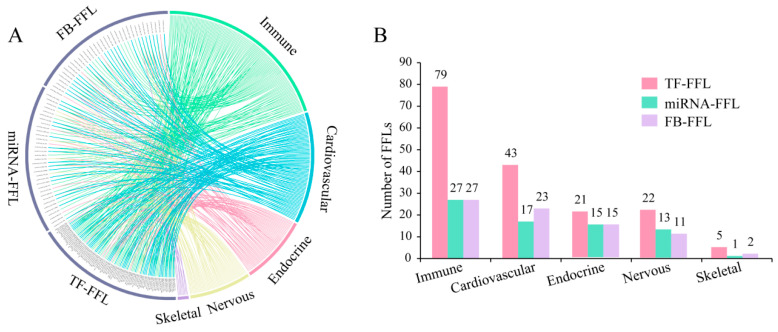
System function-related FFLs. (**A**). System function-related FFLs in five systems including immune, cardiovascular, endocrine, nervous and skeletal systems, which encoded by green, blue, red, yellow and purple, respectively. Edges connect the FFLs with the functions of body systems. (**B**). Number of dysregulated TF-FFLs (red), miRNA-FFLs (green) and FB-FFLs (purple) in microgravity for five human body systems.

**Figure 3 ijms-21-07585-f003:**
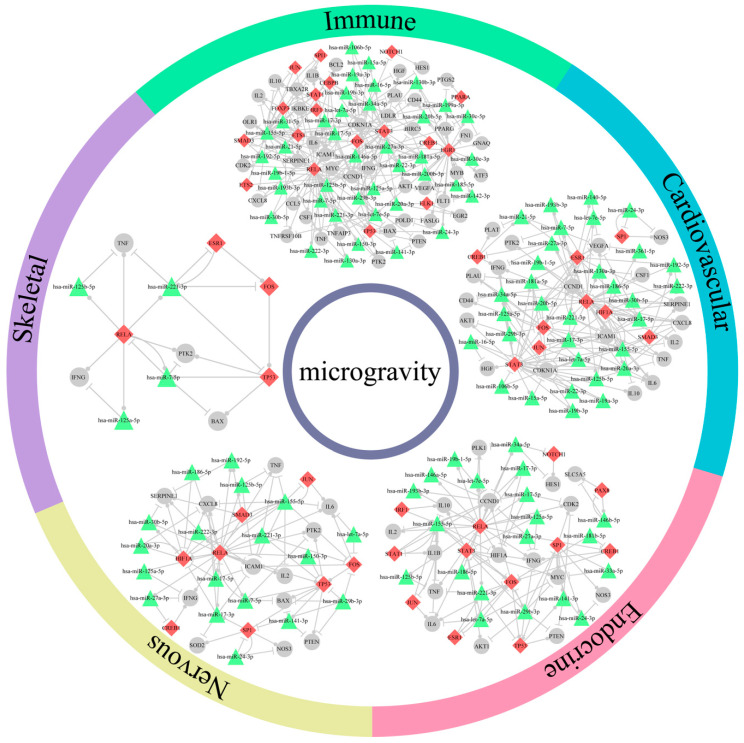
Subnetworks for five human body systems. The red diamond, green triangle and grey circular node represent TF, miRNA and gene, respectively.

**Figure 4 ijms-21-07585-f004:**
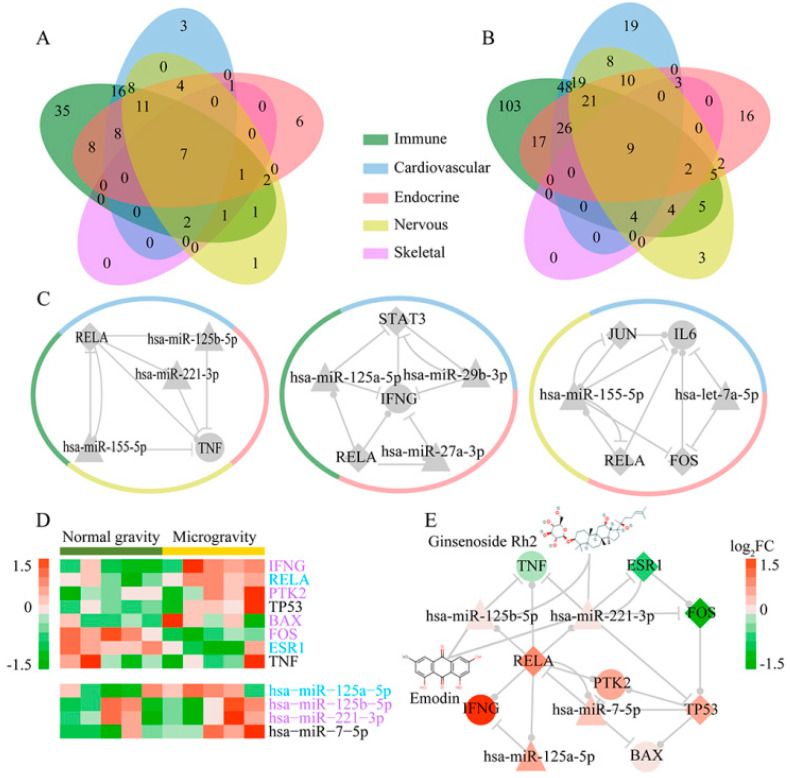
Five human body system subnetwork analysis and candidate drugs for bone loss. Venn diagram of nodes (**A**) and edges (**B**) in five subnetworks. (**C**). Modules shared by at least three subnetworks. The diamond, triangle and circular node represent TF, miRNA and gene, respectively. The colors of the circle outside the module represent corresponding systems. (**D**). Heatmap of genes and miRNAs in skeletal system subnetwork. Purple and blue text represent that the gene or miRNA functions are related to osteoblast and osteoclast, respectively. (**E**). Skeletal system subnetwork and the potential drugs reversing bone loss. Red and green represent upregulation and downregulation, respectively. The darker color represents larger |log_2_FC| value. The diamond, triangle and circular node represent TF, miRNA and gene, respectively.

**Figure 5 ijms-21-07585-f005:**
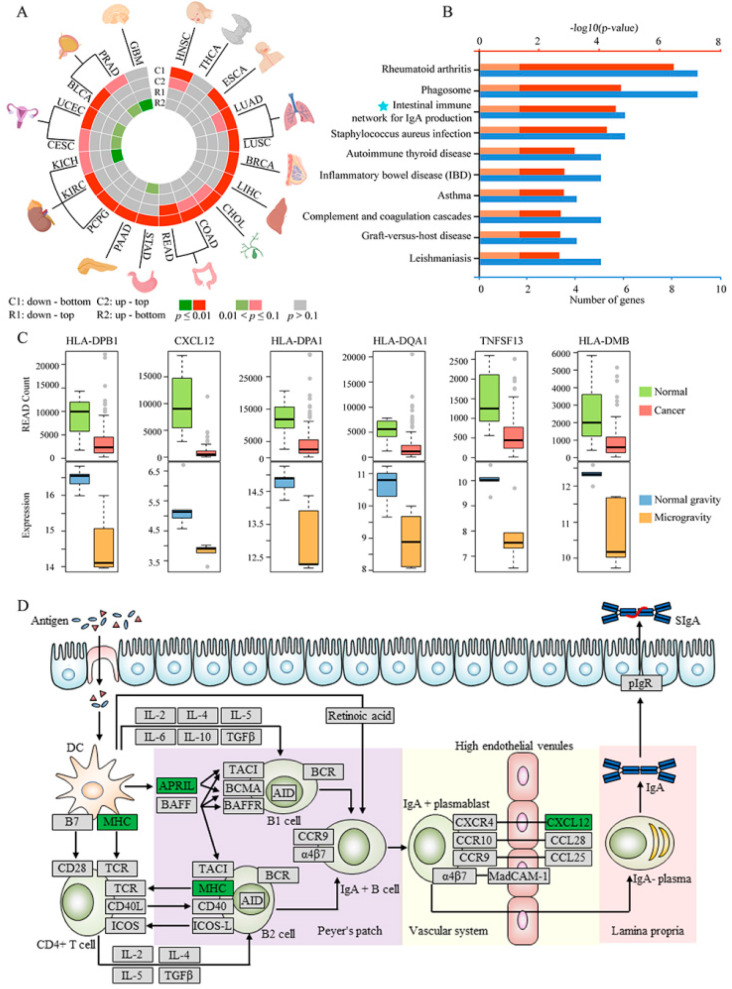
Relationships between microgravity and cancers. (**A**). Wilcoxon rank-sum test results. (**B**). Pathway enrichment analysis for 115 DE genes. The length of blue and red bar represents the number of genes and −log10(*p*-value), respectively. The pink bar represents *p*-value = 0.05. (**C**). Boxplots of the expression of six genes (*HLA-DPA1*, *HLA-DQA1*, *HLA-DPB1*, *HLA-DMB*, *CXCL12* and *TNFSF13*) in rectum adenocarcinoma (READ) samples and microgravity samples. (**D**). Intestinal immune network for IgA production pathway. The green box represents the downregulated gene enriched in the pathway.

**Table 1 ijms-21-07585-t001:** Sample size and number of differentially expressed genes for 20 cancers in the Cancer Genome Atlas (TCGA).

Cancer (Abbreviation)	#Tumor	#Normal	#Upregulated	#Downregulated
Head and neck squamous cell carcinoma (HNSC)	500	44	1162	1040
Thyroid carcinoma (THCA)	502	58	995	512
Esophageal carcinoma (ESCA)	161	11	956	956
Lung adenocarcinoma (LUAD)	533	59	1820	1221
Lung squamous cell carcinoma (LUSC)	501	49	2519	1869
Breast invasive carcinoma (BRCA)	1091	113	1557	1220
Liver hepatocellular carcinoma (LIHC)	371	50	1230	788
Cholangiocarcinoma (CHOL)	36	9	3133	2005
Colon adenocarcinoma (COAD)	478	41	1647	1327
Rectum adenocarcinoma (READ)	165	10	1624	1620
Stomach adenocarcinoma (STAD)	375	32	937	1183
Pancreatic adenocarcinoma (PAAD)	177	4	15	418
Pheochromocyt-oma and paraganglioma (PCPG)	177	3	2097	1449
Kidney renal clear cell carcinoma (KIRC)	530	72	2263	1018
Kidney chromophobe (KICH)	65	24	1630	1527
Cervical squamous cell carcinoma and endocervical adenocarcinoma (CESC)	304	3	1454	1612
Uterine corpus endometrial carcinoma (UCEC)	543	35	1961	1544
Bladder urothelial carcinoma (BLCA)	408	19	1274	1289
Prostate adenocarcinoma (PRAD)	495	52	631	914
Glioblastoma multiforme (GBM)	154	5	2987	2556

# represents “the number of”.

**Table 2 ijms-21-07585-t002:** Statistics of the regulations in the integrated TF-miRNA regulatory network.

Database	TF-Gene	TF-miRNA	MiRNA-Gene
TRANSFAC	4693	79	2013
TarBase	-	-	4158
miRTarbase	-	-	9315
TransmiR	-	3921	-
Total	4693	3930	12,052

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
