# Peer review of "Integrative Analysis of Regulatory Module Reveals Associations of Microgravity with Dysfunctions of Multi-body Systems and Tumorigenesis"

_ijms, 2020, doi:10.3390/ijms21207585_

Round 1
Reviewer 1 Report
The study submitted by Yuan et al. shows the results of an integrative analysis of miRNA expression, RNA expression and transcription factor binding in the context of microgravity and cancer. The authors did pathway enrichment analysis for gravity dependent regulated genes and miRNAs in feed-forward loops FFLs). The authors observed 160 FFLs related to so called “body systems”. Furthermore they investigated the relationships between microgravity and cancers. The authors postulated HLA-DPA1, HLA-DQA1, HLA-DPB1, HLA-DMB, TNFSF13 and CXCL12 as Rectum adenocarcinoma biomarkers possibly associated to microgravity.
Integrative analyses have significance in the determination of regulatory processes Therefore; studies that link the regulation of microRNAs and transcription factors to diseases and pathways are to be welcomed. In the present study, however, decisive factors have not been mentioned and are therefore probably disregarded. It will be difficult for the reader to evaluate the study without considering aspects such as data type (array or RNAseq) normalization, sample set size and tissue.
Major marks:
The nature of the original data processing is not described by the authors. Their differences are of outstanding importance for data comparability. The microgravity datasets are array based while the cancer datasets are RNAseq based (information taken from the original source). They differ fundamentally in sensitivity and specificity. Furthermore, the aspect of data normalization has been left out. Not all methods are suitable for an intergenic comparison. This is of importance here, since we work with ranked gene lists. Data type and normalization should be disclosed by the authors.
The central microgravity data set is array-based and has only 5 microgravity samples and 5 controls. It is therefore significantly weaker than the RNAseq cancer data sets described in table 2. Due to its size, the microgravity data set is only partially representative. A systematic comparison with data sets from other sources is therefore prone to errors. This should be addressed by the authors. The microgravity dataset is based on lymphocytes. Tissue-specific expression leads to distortions when compared with “body systems”. Accordingly this aspect should be removed or corrected for tissue specificity.
Minor marks:
2.5. Deep insight into the relationships between microgravity and cancers
P-value thresholds instead of P-values are given for HLA-DPA1 to TNFSF13
4.5. Differential expression analysis of 20 cancer types
The statistical method for determining differential expression is not specified.
Table 2
The table header and legend are not intuitive.
Figure 1
Figure 1 A is dispensable
Supplement
Supplemental tables are missing a legend and sometimes even a column header.
Reviewer 2 Report
The study by Mengqin Yuan et al. elucidates that transcriptional factors (TF) and miRNA expression profiles in human peripheral blood lymphocytes cultured in modeled microgravity.
The authors created TF and miRNA regulatory network and validated the feed-forward loops (FFLs). They determined dysregulated FFLs in modeled microgravity and found that many cancer-related genes were included. They applied this method for several human body systems and found that some TFs, such as RELA and FOS, were related to multiple systems in microgravity-induced transcriptional changes. Finally, they focused on cancer-related genes and showed that many of them were dysregulated by microgravity stimulant, suggesting that the microgravity environment provokes the tumor growth.
General comments. The analysis of transcriptional changes during microgravity is relatively rare, so the result is valuable and interesting. However, I have some concerns about the interpretation of these results. In order to further improve the work, it is advisable to consider following points. Major and minor concerns are described below.
Major points:
- As a main sensor of the gravity in vivo, the vestibular system is well-known and broadly takes part in the physiological reactions to gravitational changes (Morita et al, J PHYSIOL SCI, 2020). While in this experimentin vitro, how does the HPBLs detect the gravitational changes? Do they express any mechanosensory receptors?
- Is the mechanism of detecting gravity shared with whole body cells? The authors analyzed and concluded the transcriptional changes in multiple human body systems only from HPBLs data, but I concerned that the reaction to microgravity would be different in each type of cells.
Minor points:
1 Describing the summary of the method in microgravity experiments would make this manuscript more reader-friendly.
Round 2
Reviewer 1 Report
All suggestions were adopted by the authors. Therefore I recommend the publication in the present form.
Reviewer 2 Report
The authors have basically addressed all of my concerns.
I think this paper can be accepted.